# Deep Learning without Weight Transport

**Mohamed Akrout**
University of Toronto, Triage

**Collin Wilson**
University of Toronto

**Peter C. Humphreys**
DeepMind

**Timothy Lillicrap**
DeepMind, University College London

**Douglas Tweed**
University of Toronto, York University

## Abstract

Current algorithms for deep learning probably cannot run in the brain because they rely on *weight transport*, where forward-path neurons transmit their synaptic weights to a feedback path, in a way that is likely impossible biologically. An algorithm called feedback alignment achieves deep learning without weight transport by using random feedback weights, but it performs poorly on hard visual-recognition tasks. Here we describe two mechanisms — a neural circuit called a *weight mirror* and a modification of an algorithm proposed by Kolen and Pollack in 1994 — both of which let the feedback path learn appropriate synaptic weights quickly and accurately even in large networks, without weight transport or complex wiring. Tested on the ImageNet visual-recognition task, these mechanisms learn almost as well as backprop (the standard algorithm of deep learning, which uses weight transport) and they outperform feedback alignment and another, more-recent transport-free algorithm, the sign-symmetry method.

## 1 Introduction

The algorithms of deep learning were devised to run on computers, yet in many ways they seem suitable for brains as well; for instance, they use multilayer networks of processing units, each with many inputs and a single output, like networks of neurons. But current algorithms can't quite work in the brain because they rely on the error-backpropagation algorithm, or backprop, which uses *weight transport*: each unit multiplies its incoming signals by numbers called weights, and some units transmit their weights to other units. In the brain, it is the synapses that perform this weighting, but there is no known pathway by which they can transmit their weights to other neurons or to other synapses in the same neuron [1, 2].

Lillicrap et al. [3] offered a solution in the form of feedback alignment, a mechanism that lets deep networks learn without weight transport, and they reported good results on several tasks. But Bartunov et al. [4] and Moskovitz et al. [5] have found that feedback alignment does not scale to hard visual recognition problems such as ImageNet [6].

Xiao et al. [7] achieved good performance on ImageNet using a *sign-symmetry* algorithm in which only the signs of the forward and feedback weights, not necessarily their values, must correspond, and they suggested a mechanism by which that correspondence might be set up during brain development. Krotov and Hopfield [8] and Guerguiev et al. [9] have explored other approaches to deep learning without weight transport, but so far only in smaller networks and tasks.

Here we propose two different approaches that learn ImageNet about as well as backprop does, with no need to initialize forward and feedback matrices so their signs agree. We describe a circuit called a *weight mirror* and a version of an algorithm proposed by Kolen and Pollack in 1994 [10], both of which let initially random feedback weights learn appropriate values without weight transport.

There are of course other questions about the biological implications of deep-learning algorithms, some of which we touch on in Appendix C, but in this paper our main concern is with weight transport.

## 2 The weight-transport problem

In a typical deep-learning network, some signals flow along a *forward path* through multiple layers of processing units from the input layer to the output, while other signals flow back from the output layer along a *feedback path*. Forward-path signals perform inference (e.g. they try to infer what objects are depicted in a visual input) while the feedback path conveys error signals that guide learning. In the forward path, signals flow according to the equation

$$\mathbf{y}_{l+1} = \phi(\mathbf{W}_{l+1} \, \mathbf{y}_l + \mathbf{b}_{l+1}) \tag{1}$$

Here $\mathbf{y}_l$ is the output signal of layer $l$, i.e. a vector whose $i$-th element is the activity of unit $i$ in layer $l$. Equation 1 shows how the next layer $l+1$ processes its input $\mathbf{y}_l$: it multiplies $\mathbf{y}_l$ by the forward weight matrix $\mathbf{W}_{l+1}$, adds a bias vector $\mathbf{b}_{l+1}$, and puts the sum through an activation function $\phi$. Interpreted as parts of a real neuronal network in the brain, the $\mathbf{y}$'s might be vectors of neuronal firing rates, or some function of those rates, $\mathbf{W}_{l+1}$ might be arrays of synaptic weights, and $\mathbf{b}_{l+1}$ and $\phi$ bias currents and nonlinearities in the neurons.

In the feedback path, error signals $\boldsymbol{\delta}$ flow through the network from its output layer according to the error-backpropagation [11] or backprop equation:

$$\boldsymbol{\delta}_l = \phi'(\mathbf{y}_l) \, \mathbf{W}_{l+1}^T \, \boldsymbol{\delta}_{l+1} \tag{2}$$

Here $\phi'$ is the derivative of the activation function $\phi$ from equation (1), which can be computed from $\mathbf{y}_l$. So feedback signals pass layer by layer through weights $\mathbf{W}_l^T$. Interpreted as a structure in the brain, the feedback path might be another set of neurons, distinct from those in the forward path, or the same set of neurons might carry inference signals in one direction and errors in the other [12, 13].

Either way, we have the problem that the same weight matrix $\mathbf{W}_l$ appears in the forward equation (1) and then again, transposed, in the feedback equation (2), whereas in the brain, the synapses in the forward and feedback paths are physically distinct, with no known way to coordinate themselves so one set is always the transpose of the other [1, 2].

## 3 Feedback alignment

In feedback alignment, the problem is avoided by replacing the transposed $\mathbf{W}_l$'s in the feedback path by random, fixed (non-learning) weight matrices $\mathbf{B}_l$,

$$\boldsymbol{\delta}_l = \phi'(\boldsymbol{y}_l) \, \mathbf{B}_{l+1} \, \boldsymbol{\delta}_{l+1} \tag{3}$$

These feedback signals $\boldsymbol{\delta}$ drive learning in the forward weights $\mathbf{W}$ by the rule

$$\Delta\mathbf{W}_{l+1} = -\eta_W \, \boldsymbol{\delta}_{l+1} \, \mathbf{y}_l^T \tag{4}$$

where $\eta_W$ is a learning-rate factor. As shown in [3], equations (1), (3), and (4) together drive the forward matrices $\mathbf{W}_l$ to become roughly proportional to transposes of the feedback matrices $\mathbf{B}_l$. That rough transposition makes equation (3) similar enough to the backprop equation (2) that the network can learn simple tasks as well as backprop does.

Can feedback alignment be augmented to handle harder tasks? One approach is to adjust the feedback weights $\mathbf{B}_l$ as well as the forward weights $\mathbf{W}_l$, to improve their agreement. Here we show two mechanisms by which that adjustment can be achieved quickly and accurately in large networks without weight transport.

## 4 Weight mirrors

### 4.1 Learning the transpose

The aim here is to adjust an initially random matrix $\mathbf{B}$ so it becomes proportional to the transpose of another matrix $\mathbf{W}$ without weight transport, i.e. given only the input and output vectors $\mathbf{x}$ and

$\mathbf{y} = \mathbf{W}\mathbf{x}$ (for this explanation, we neglect the activation function $\phi$). We observe that $E\left[\mathbf{x}\,\mathbf{y}^T\right] = E\left[\mathbf{x}\,\mathbf{x}^T\mathbf{W}^T\right] = E\left[\mathbf{x}\,\mathbf{x}^T\right]\mathbf{W}^T$. In the simplest case, if the elements of $\mathbf{x}$ are independent and zero-mean with equal variance, $\sigma^2$, it follows that $E\left[\mathbf{x}\,\mathbf{y}^T\right] = \sigma^2\mathbf{W}^T$. Therefore we can push $\mathbf{B}$ steadily in the direction $\sigma^2\mathbf{W}$ using this *transposing rule*,

$$\Delta\mathbf{B} = \eta_B\,\mathbf{x}\,\mathbf{y}^T \tag{5}$$

So $\mathbf{B}$ integrates a signal that is proportional to $\mathbf{W}^T$ on average. Over time, this integration may cause the matrix norm $\|\mathbf{B}\|$ to increase, but if we add a mechanism to keep the norm small — such as weight decay or synaptic scaling [14–16] — then the initial, random values in $\mathbf{B}$ shrink away, and $\mathbf{B}$ converges to a scalar multiple of $\mathbf{W}^T$ (see Appendix A for an account of this learning rule in terms of gradient descent).

## 4.2   A circuit for transposition

Figure 1 shows one way the learning rule (5) might be implemented in a neural network. This network alternates between two modes: an *engaged mode*, where it receives sensory inputs and adjusts its forward weights to improve its inference, and a *mirror mode*, where its neurons discharge noisily and adjust the feedback weights so they mimic the forward ones. Biologically, these two modes may correspond to wakefulness and sleep, or simply to practicing a task and then setting it aside for a moment.

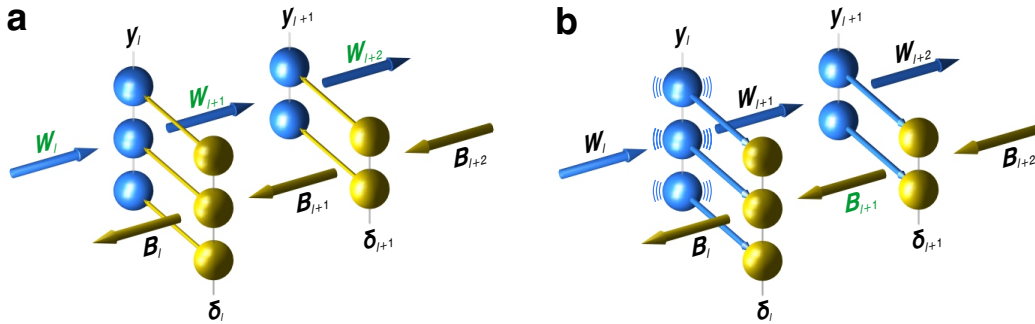

Figure 1: Network modes for weight mirroring. Both panels show the same two-layer section of a network. In both modes, the three neurons in layer $l$ of the forward path (■) send their output signal $\mathbf{y}_l$ through the weight array $\mathbf{W}_{l+1}$ (and other processing shown in equation (1)) to yield the next-layer signal $\mathbf{y}_{l+1}$. And in the feedback path (■), the two neurons in layer $l + 1$ send their signal $\boldsymbol{\delta}_{l+1}$ through weight array $\mathbf{B}_{l+1}$ to yield $\boldsymbol{\delta}_l$, as in (3). The figure omits the biases $\mathbf{b}$, nonlinearities $\phi$, and, in the top panel, the projections that convey $\mathbf{y}_l$ to the $\boldsymbol{\delta}_l$ cells, allowing them to compute the factor $\phi'(\mathbf{y}_l)$ in equation (3). **a)** In *engaged mode*, cross-projections (—) convey the feedback signals $\boldsymbol{\delta}$ to the forward-path cells, so they can adjust the forward weights $\mathbf{W}$ using learning rule (4). **b)** In *mirror mode*, one layer of forward cells, say layer $l$, fires noisily. Its signal $\mathbf{y}_l$ still passes through $\mathbf{W}_{l+1}$ to yield $\mathbf{y}_{l+1}$, but now the blue cross-projections (—) control firing in the feedback path, so $\boldsymbol{\delta}_l = \mathbf{y}_l$ and $\boldsymbol{\delta}_{l+1} = \mathbf{y}_{l+1}$, and the $\boldsymbol{\delta}_l$ neurons adjust the feedback weights $\mathbf{B}_{l+1}$ using learning rule (7). We call the circuit $\mathbf{y}_l,\ \mathbf{y}_{l+1},\ \boldsymbol{\delta}_{l+1},\ \boldsymbol{\delta}_l$ a *weight mirror* because it makes the weight array $\mathbf{B}_{l+1}$ resemble $\mathbf{W}_{l+1}^T$.

In mirror mode, the forward-path neurons in each layer $l$, carrying the signal $\mathbf{y}_l$, project strongly to layer $l$ of the feedback path — strongly enough that each signal $\boldsymbol{\delta}_l$ of the feedback path faithfully mimics $\mathbf{y}_l$, i.e.

$$\boldsymbol{\delta}_l = \mathbf{y}_l \tag{6}$$

Also in mirror mode, those forward-path signals $\mathbf{y}_l$ are noisy. Multiple layers may fire at once, but the process is simpler to explain in the case where they take turns, with just one layer $l$ driving forward-path activity at any one time. In that case, all the cells of layer $l$ fire randomly and independently, so their output signal $\mathbf{y}_l$ has zero-mean and equal variance $\sigma^2$. That signal passes through forward weight matrix $\mathbf{W}_{l+1}$ and activation function $\phi$ to yield $\mathbf{y}_{l+1} = \phi(\mathbf{W}_{l+1}\,\mathbf{y}_l + b_l)$. By equation (6),

signals $\mathbf{y}_l$ and $\mathbf{y}_{l+1}$ are transmitted to the feedback path. Then the layer-$l$ feedback cells adjust their weights $\mathbf{B}_{l+1}$ by Hebbian learning,

$$\Delta \mathbf{B}_{l+1} = \eta_B \ \boldsymbol{\delta}_l \ \boldsymbol{\delta}_{l+1}^T \tag{7}$$

This circuitry and learning rule together constitute the weight mirror.

### 4.3 Why it works

To see that (7) approximates the transposing rule (5), notice first that

$$\boldsymbol{\delta}_l \ \boldsymbol{\delta}_{l+1}^T = \mathbf{y}_l \ \mathbf{y}_{l+1}^T = \mathbf{y}_l \ \phi(\mathbf{W}_{l+1} \ \mathbf{y}_l + \mathbf{b}_{l+1})^T \tag{8}$$

We will assume, for now, that the variance $\sigma^2$ of $\mathbf{y}_l$ is small enough that $\mathbf{W}_{l+1} \ \mathbf{y}_l + \mathbf{b}_{l+1}$ stays in a roughly affine range of $\phi$, and that the diagonal elements of the derivative matrix $\phi'(\mathbf{b}_{l+1})$ are all roughly similar to each other, so the matrix is approximately of the form $\phi'_s \mathbf{I}$, where $\phi'_s$ is a positive scalar and $\mathbf{I}$ is the identity. Then

$$\begin{aligned}\phi(\mathbf{W}_{l+1} \ \mathbf{y}_l + \mathbf{b}_{l+1}) &\approx \phi'(\mathbf{b}_{l+1}) \ \mathbf{W}_{l+1} \ \mathbf{y}_l + \phi(\mathbf{b}_{l+1}) \\ &\approx \phi'_s \ \mathbf{W}_{l+1} \ \mathbf{y}_l + \phi(\mathbf{b}_{l+1})\end{aligned} \tag{9}$$

Therefore

$$\boldsymbol{\delta}_l \ \boldsymbol{\delta}_{l+1}^T \approx \mathbf{y}_l \big[ \ \mathbf{y}_l^T \ \mathbf{W}_{l+1}^T \ \phi'_s + \phi(\mathbf{b}_{l+1})^T \ \big] \tag{10}$$

and so

$$\begin{aligned}E\big[\Delta \mathbf{B}_{l+1}\big] &\approx \eta_B \left( E\big[\mathbf{y}_l \mathbf{y}_l^T\big] \mathbf{W}_{l+1}^T \phi'_s + E\big[\mathbf{y}_l\big] \phi(\mathbf{b}_{l+1})^T \right) \\ &= \eta_B \ E\big[\mathbf{y}_l \mathbf{y}_l^T\big] \mathbf{W}_{l+1}^T \phi'_s \\ &= \eta_B \ \sigma^2 \phi'_s \mathbf{W}_{l+1}^T\end{aligned} \tag{11}$$

Hence the weight matrix $\mathbf{B}_{l+1}$ integrates a teaching signal (7) which approximates, on average, a positive scalar multiple of $\mathbf{W}_{l+1}^T$. As in (5), this integration may drive up the matrix norm $\|\mathbf{B}_{l+1}\|$, but if we add a mechanism such as weight decay to keep the norm small [15, 16] then $\mathbf{B}_{l+1}$ evolves toward a reasonable-sized positive multiple of $\mathbf{W}_{l+1}^T$.

We get a stronger result if we suppose that neurons are capable of *bias-blocking* — of closing off their bias currents when in mirror mode, or preventing their influence on the axon hillock. Then

$$E\big[\Delta \mathbf{B}_{l+1}\big] \approx \eta_B \ \sigma^2 \phi'(\mathbf{0}) \mathbf{W}_{l+1}^T \tag{12}$$

So again, $\mathbf{B}_{l+1}$ comes to approximate a positive scalar multiple of $\mathbf{W}_{l+1}^T$, so long as $\phi$ has a positive derivative around 0, but we no longer need to assume that $\phi'(\mathbf{b}_{l+1}) \approx \phi'_s \mathbf{I}$.

In one respect the weight mirror resembles difference target propagation [4], because both mechanisms shape the feedback path layer by layer, but target propagation learns layer-wise autoencoders (though see [17]), and uses feedback weights to propagate targets rather than gradients.

## 5 The Kolen-Pollack algorithm

### 5.1 Convergence through weight decay

Kolen and Pollack [10] observed that we don't have to transport *weights* if we can transport *changes* in weights. Consider two synapses, $W$ in the forward path and $B$ in the feedback path (written without boldface because for now we are considering individual synapses, not matrices). Suppose $W$ and $B$ are initially unequal, but at each time step $t$ they undergo identical adjustments $A(t)$ and apply identical weight-decay factors $\lambda$, so

$$\Delta W(t) = A(t) - \lambda W(t) \tag{13}$$

and

$$\Delta B(t) = A(t) - \lambda B(t) \tag{14}$$

Then $W(t+1) - B(t+1) = W(t) + \Delta W(t) - B(t) - \Delta B(t) = W(t) - B(t) - \lambda[W(t) - B(t)] = (1 - \lambda)[W(t) - B(t)] = (1 - \lambda)^{t+1}[W(0) - B(0)]$, and so with time, if $0 < \lambda < 1$, $W$ and $B$ will converge.

But biologically, it is no more feasible to transport weight changes than weights, and Kolen and Pollack do not say how their algorithm might run in the brain. Their flow diagram (Figure 2 in their paper) is not at all biological: it shows weight changes being calculated at one locus and then traveling to distinct synapses in the forward and feedback paths. In the brain, changes to different synapses are almost certainly calculated separately, within the synapses themselves. But it *is* possible to implement Kolen and Pollack's method in a network without transporting weights or weight changes.

## 5.2 A circuit for Kolen-Pollack learning

The standard, forward-path learning rule (4) says that the matrix $\mathbf{W}_{l+1}$ adjusts itself based on a product of its input vector $\mathbf{y}_l$ and a teaching vector $\boldsymbol{\delta}_{l+1}$. More specifically, each synapse $W_{l+1,ij}$ adjusts itself based on its own scalar input $y_{l,j}$ and the scalar teaching signal $\delta_{l+1,i}$ sent to its neuron from the feedback path.

We propose a reciprocal arrangement, where synapses in the feedback path adjust themselves based on their own inputs and cell-specific, scalar teaching signals from the forward path,

$$\Delta \mathbf{B}_{l+1} = -\eta \, \mathbf{y}_l \, \boldsymbol{\delta}_{l+1}^T \tag{15}$$

If learning rates and weight decay agree in the forward and feedback paths, we get

$$\Delta \mathbf{W}_{l+1} = -\eta_W \, \boldsymbol{\delta}_{l+1} \, \mathbf{y}_l^T - \lambda \, \mathbf{W}_{l+1} \tag{16}$$

and

$$\Delta \mathbf{B}_{l+1} = -\eta_W \, \mathbf{y}_l \, \boldsymbol{\delta}_{l+1}^T - \lambda \, \mathbf{B}_{l+1} \tag{17}$$

i.e.

$$\Delta \mathbf{B}_{l+1}^T = -\eta_W \, \boldsymbol{\delta}_{l+1} \, \mathbf{y}_l^T - \lambda \, \mathbf{B}_{l+1}^T \tag{18}$$

In this network (drawn in Figure 2), the only variables transmitted between cells are the activity vectors $\mathbf{y}_l$ and $\boldsymbol{\delta}_{l+1}$, and each synapse computes its own adjustment locally, but (16) and (18) have the form of the Kolen-Pollack equations (13) and (14), and therefore the forward and feedback weight matrices converge to transposes of each other.

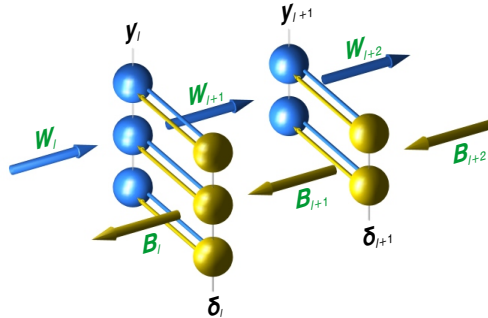

Figure 2: Reciprocal network for Kolen-Pollack learning. There is a single mode of operation. Gold-colored cross-projections (—) convey feedback signals $\boldsymbol{\delta}$ to forward-path cells, so they can adjust the forward weights $\mathbf{W}$ using learning rule (16). Blue cross-projections (—) convey the signals $\mathbf{y}$ to the feedback cells, so they can adjust the feedback weights $\mathbf{B}$ using (17).

We have released a Python version of the proprietary TensorFlow/TPU code for the weight mirror and the KP reciprocal network that we used in our tests; see `github.com/makrout/Deep-Learning-without-Weight-Transport`.

## 6 Experiments

We compared our weight-mirror and Kolen-Pollack networks to backprop, plain feedback alignment, and the sign-symmetry method [5, 7]. For easier comparison with recent papers on biologically-motivated algorithms [4, 5, 7], we used the same types of networks they did, with convolution [18],

batch normalization (BatchNorm) [19], and rectified linear units (ReLUs) without bias-blocking. In most experiments, we used a ResNet block variant where signals were normalized by BatchNorm after the ReLU nonlinearity, rather than before (see Appendix D.3). More brain-like implementations would have to replace BatchNorm with some kind of synaptic scaling [15, 16], ReLU with a bounded function such as rectified tanh, and convolution with non-weight-sharing local connections.

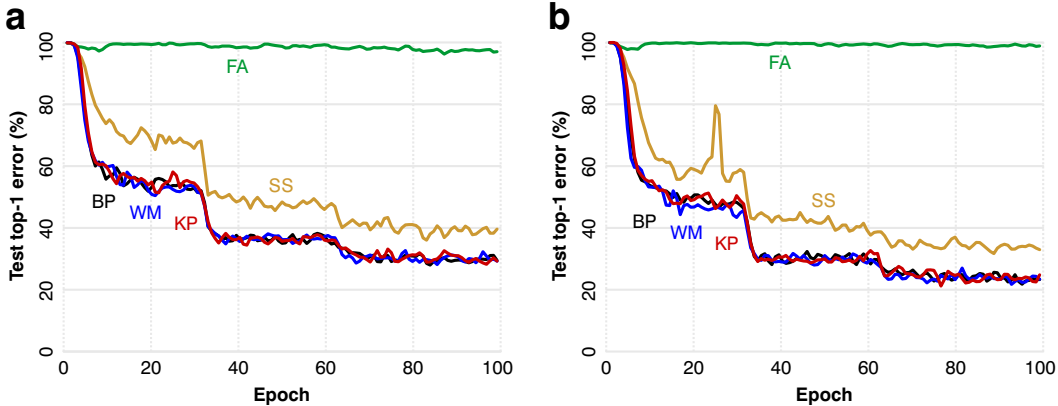

Figure 3: ImageNet results. **a)** With ResNet-18 architecture, the weight-mirror network (— WM) and Kolen-Pollack (— KP) outperformed plain feedback alignment (— FA) and the sign-symmetry algorithm (— SS), and nearly matched backprop (— BP). **b)** With the larger ResNet-50 architecture, results were similar.

Run on the ImageNet visual-recognition task [6] with the ResNet-18 network (Figure 3a), weight mirrors managed a final top-1 test error of $30.2(7)\%$, and Kolen-Pollack reached $29.2(4)\%$, versus $97.4(2)\%$ for plain feedback alignment, $39.2(4)\%$ for sign-symmetry, and $30.1(4)\%$ for backprop. With ResNet-50 (Figure 3b), the scores were: weight mirrors $23.4(5)\%$, Kolen-Pollack $23.9(7)\%$, feedback alignment $98.9(1)\%$, sign-symmetry $33.8(3)\%$, and backprop $22.9(4)\%$. (Digits in parentheses are standard errors).

Sign-symmetry did better in other experiments where batch normalization was applied *before* the ReLU nonlinearity. In those runs, it achieved top-1 test errors of $37.8(4)\%$ with ResNet-18 (close to the $37.91\%$ reported in [7] for the same network) and $32.6(6)\%$ with ResNet-50 (see Appendix D.1 for details of our hyperparameter selection, and Appendix D.3 for a figure of the best result attained by sign-symmetry on our tests). The same change in BatchNorm made little difference to the other four methods — backprop, feedback alignment, Kolen-Pollack, and the weight mirror.

Weight mirroring kept the forward and feedback matrices in agreement throughout training, as shown in Figure 4. One way to measure this agreement is by matrix angles: in each layer of the networks, we took the feedback matrix $\mathbf{B}_l$ and the transpose of the forward matrix, $\mathbf{W}_l^T$, and reshaped them into vectors. With backprop, the angle between those vectors was of course always 0. With weight mirrors (Figure 4a), the angle stayed $< 12°$ in all layers, and $< 6°$ later in the run for all layers except the final one. That final layer was fully connected, and therefore its $\mathbf{W}_l$ received more inputs than those of the other, convolutional layers, making its $\mathbf{W}_l^T$ harder to deduce. For closer alignment, we would have needed longer mirroring with more examples.

The matrix angles grew between epochs 2 and 10 and then held steady at relatively high levels till epoch 32 because during this period the learning rate $\eta_W$ was large (see Appendix D.1), and mirroring didn't keep the $\mathbf{B}_l$'s matched to the fast-changing $\mathbf{W}_l^T$'s. That problem could also have been solved with more mirroring, but it did no harm because at epoch 32, $\eta_W$ shrank by 90%, and from then on, the $\mathbf{B}_l$'s and $\mathbf{W}_l^T$'s stayed better aligned.

We also computed the $\boldsymbol{\delta}$ angles between the feedback vectors $\boldsymbol{\delta}_l$ computed by the weight-mirror network (using $\mathbf{B}_l$'s) and those that would have been computed by backprop (using $\mathbf{W}_l^T$'s). Weight mirrors kept these angles $< 25°$ in all layers (Figure 4b), with worse alignment farther upstream, because $\boldsymbol{\delta}$ angles depend on the accumulated small discrepancies between all the $\mathbf{B}_l$'s and $\mathbf{W}_l^T$'s in all downstream layers.

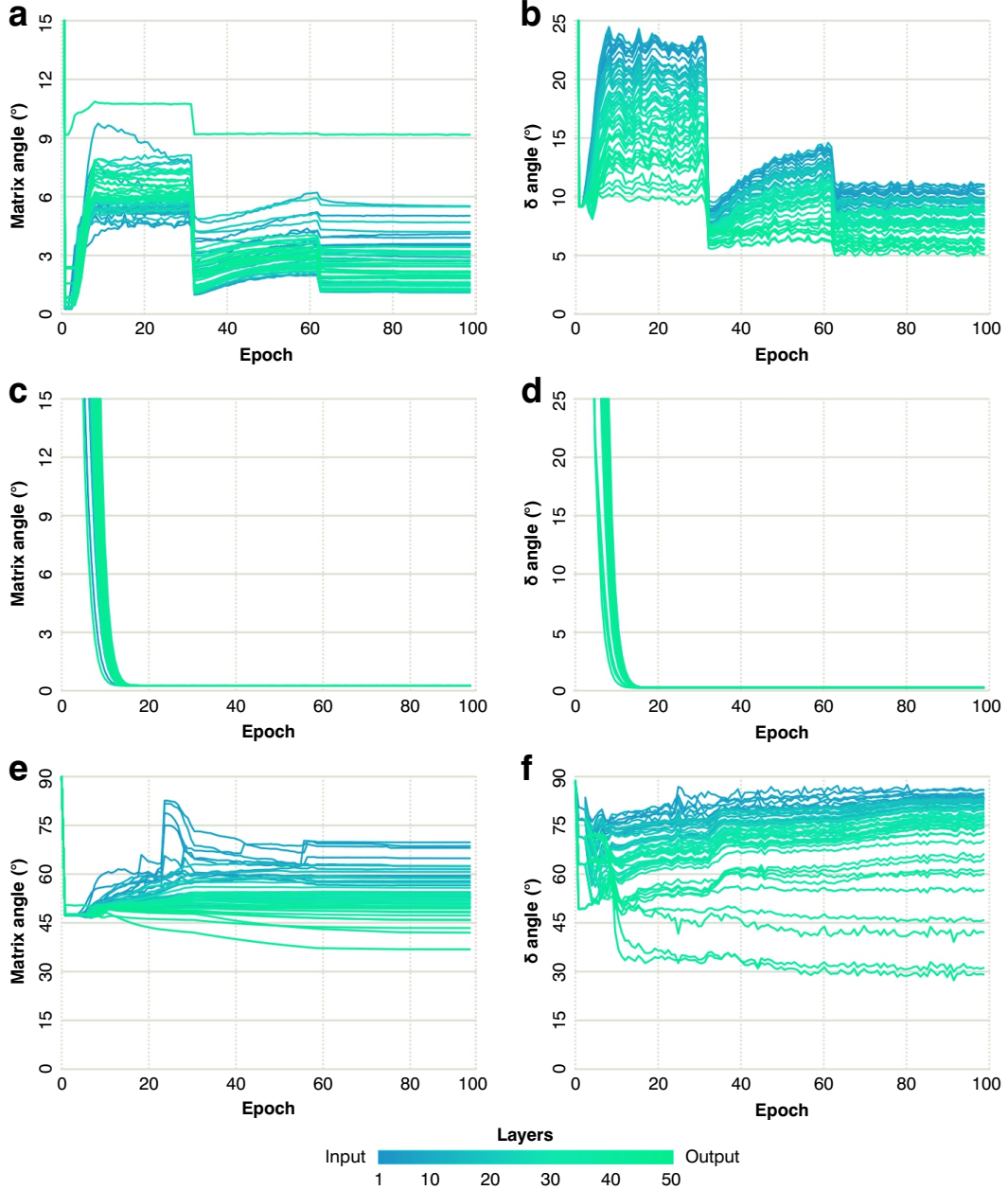

Figure 4: Agreement of forward and feedback matrices in the ResNet-50 from Figure 3b. **a)** Weight mirrors kept the angles between the matrices $\mathbf{B}_l$ and $\mathbf{W}_l^T$ small in all layers, from the input layer (—) to the output (—). **b)** Feedback vectors $\boldsymbol{\delta}_l$ computed by the weight-mirror network were also well aligned with those that would have been computed by backprop. **c, d)** The Kolen-Pollack network kept the matrix and $\boldsymbol{\delta}$ angles even smaller. **e, f)** The sign-symmetry method was less accurate.

The Kolen-Pollack network was even more accurate, bringing the matrix and $\boldsymbol{\delta}$ angles to near zero within 20 epochs and holding them there, as shown in Figures 4c and 4d.

The sign-symmetry method aligned matrices and $\boldsymbol{\delta}$'s less accurately (Figures 4e and 4f), while with feedback alignment (not shown), both angles stayed > 80° for most layers in both the ResNet-18 and ResNet-50 architectures.

## 7 Discussion

Both the weight mirror and the Kolen-Pollack network outperformed feedback alignment and the sign-symmetry algorithm, and both kept pace, at least roughly, with backprop. Kolen-Pollack has some advantages over weight mirrors, as it doesn't call for separate modes of operation and needn't proceed layer by layer. Conversely, weight mirrors don't need sensory input but learn from noise, so they could tune feedback paths in sleep or in utero. And while KP kept matrix and $\delta$ angles smaller than WM did in Figure 4, that may not be the case in all learning tasks. With KP, the matrix $\mathbf{B}$ converges to $\mathbf{W}^T$ at a rate that depends on $\lambda$, the weight-decay factor in equation (17). A big $\lambda$ speeds up alignment, but may hamper learning, and at present we have no proof that a good balance can always be found between $\lambda$ and learning rate $\eta_W$. In this respect, WM may be more versatile than KP, because if mirroring ever fails to yield small enough angles, we can simply do more mirroring, e.g. in sleep. More tests are needed to assess the two mechanisms' aptitude for different tasks, their sensitivity to hyperparameters, and their effectiveness in non-convolutional networks and other architectures.

Both methods may have applications outside biology, because the brain is not the only computing device that lacks weight transport. Abstractly, the issue is that the brain represents information in two different forms: some is coded in action potentials, which are energetically expensive but rapidly transmissible to other parts of the brain, while other information is stored in synaptic weights, which are cheap and compact but localized — they influence the transmissible signals but are not themselves transmitted. Similar issues arise in certain kinds of technology, such as application-specific integrated circuits (ASICs). Here as in the brain, mechanisms like weight mirroring and Kolen-Pollack could allow forward and feedback weights to live locally, saving time and energy [20–22].

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
