[Supplementary Material]

## Appendices

## A  The transposing rule as gradient descent

The learning rule (5) can be expressed as a form of gradient descent,

$$\Delta \mathbf{B} = \eta \, \mathbf{x} \, \mathbf{y}^T = -\eta \, \frac{\partial f}{\partial \mathbf{B}} \tag{19}$$

where

$$f(\mathbf{x}, \mathbf{y}, \mathbf{B}) = -\mathbf{x}^T \mathbf{B} \mathbf{y} = -\sum_{i,j} x_i \, y_j \, B_{ij} \tag{20}$$

This function $f$ is not a *loss* or *objective* function, as it has no minimum for any fixed, non-zero $\mathbf{x}$ and $\mathbf{y}$, and neither is it a quantity we would *wish* to minimize, because it can be pushed farther and farther below zero by making $\mathbf{B}$ larger and larger. But if we combine (5) with weight decay

$$\Delta \mathbf{B} = \eta_B \, \mathbf{x} \, \mathbf{y}^T - \lambda_{WM} \, \mathbf{B} \tag{21}$$

then we do descend the gradient of a loss

$$\mathcal{L} = f + \frac{\lambda_{WM}}{2 \, \eta_B} \, \|\mathbf{B}\|^2 \tag{22}$$

## B  Computational costs

Weight mirroring is slightly more expensive computationally than is Kolen-Pollack learning. Suppose layers $l$ and $l + 1$ of a learning network are fully connected, with $n_l$ and $n_{l+1}$ forward units (and the same numbers of feedback units if they are separate from the forward ones), and let $n = \min(n_l, n_{l+1})$. Then for each training example, KP does $n + 4n_l n_{l+1}$ flops to adjust $\mathbf{B}_{l+1}$ using equation (17). WM does the same number to adjust $\mathbf{B}_{l+1}$ using equation (21), but WM also has to generate a random vector $y_l$ and then perform about $2n_l n_{l+1}$ flops to compute $y_{l+1}$ from $y_l$ using equation (1), whereas KP uses the same $y_l$ and $y_{l+1}$ that train the forward matrices.

## C  Biological interpretations

The variables in the weight-mirror and Kolen-Pollack equations can be interpreted physically in several different ways. Here we describe some issues and options:

### C.1  Distinct feedback neurons?

Figures 1 and 2 show learning networks where inference signals $\mathbf{y}$ and error signals $\boldsymbol{\delta}$ are carried by distinct sets of neurons. But their equations work just as well if the same neurons carry inference signals in one direction and errors in the other.

If the forward and feedback paths are *distinct* sets of neurons, then our proposed methods call for a one-to-one pairing, connecting each forward-path cell with a partner cell in the feedback path. Such connections might arise during development. We know that precise and consistent neuronal wiring is found in simple organisms such as *C. elegans* [23] and in the compound eyes of insects [24], while in the cerebella of at least some mammalian species each Purkinje cell connects with exactly one appropriate climbing fiber [25] . Alternatively, something less than strict one-to-one wiring may suffice for effective learning, and may itself be learned.

Getting a one-to-one correspondence is of course trivial if the *same* neurons make up the forward and feedback paths, though then we face the new problem of signal segregation — explaining how signals $\mathbf{y}$ and $\delta$ can flow through the same cells without interfering. Some possibilities are that neurons segregate $\mathbf{y}$ and $\delta$ by conveying them with different intracellular messengers or computing them in different parts of the cell [26] or by multiplexing [27], or cells may take turns carrying one or the other signal.

## C.2 Zero-mean signals

In equation (11), we provided a rationale for our learning rule (7), but to do it we had to assume that the signal $\mathbf{y}_l$ had a mean of zero. That assumption is awkward if we interpret the signals in our equations as firing rates of neurons, because neurons can't have negative rates, and so can't have zero means except by staying utterly silent. But we can drop the zero-mean requirement if we suppose that neurons convey positive and negative values by modulating about a baseline rate $\beta$. For instance, we might have

$$\mathbf{y}_{l+1} = \phi\big(\mathbf{W}_{l+1}(\mathbf{y}_l - \beta) + \mathbf{b}_{l+1}\big) \tag{23}$$

where $\phi$ is a non-negative activation function and $\mathbf{y}_l - \beta$ means that the same scalar $\beta$ is subtracted from each element of the signal vector $\mathbf{y}_l$. In mirror mode, $\mathbf{y}_l$ might then fire noisily with a mean of $\beta$ rather than 0. We could employ a variant of bias-blocking in which $\mathbf{b}_{l+1}$ is set to $b^-$, where $b^-$ is a *default bias* chosen so that $\phi(b^-) = \beta$ and $\phi'(b^-) > 0$. And we could adjust $\mathbf{B}_{l+1}$ by the rule

$$\Delta \mathbf{B}_{l+1} = \eta_B \, (\boldsymbol{\delta}_l - \beta)(\boldsymbol{\delta}_{l+1} - \beta)^T \tag{24}$$

The baseline $\beta$ might be built into neurons by the genome or it might be estimated locally, for example by taking the average of the firing rates over a period of mirroring, as we did in our experiments.

Another way to get positive and negative signals in the brain is to think of each processing unit not as a single neuron but as a group of cells acting in push-pull, some carrying positive signals and others negative [28]. Both mechanisms — baselines and push-pull — operate in the brain, for instance in the vestibulo-ocular reflex [29].

## C.3 Multipurpose projections?

To avoid clutter, Figure 1a omitted the cross-projections which convey $\mathbf{y}_l$ to the feedback cells, allowing them to compute the factor $\phi'(\mathbf{y}_l)$ in equation (3). Figure 1b does show cross-projections from forward to feedback cells, carrying the same signal, $\mathbf{y}_l$, but having a different effect on the target cells, setting $\boldsymbol{\delta}_l = \mathbf{y}_l$. We may interpret these two sets of projections — the ones omitted from Figure 1a and the ones drawn as thin blue arrows in 1b — as two distinct sets of axons carrying the same signal, or as a single set of axons whose effects on their targets differ in the two modes. Maybe these axons form two types of synapses, some onto ionotropic receptors and some onto metabotropic, or maybe some switch in intracellular signaling within the feedback cells makes them respond differently to identical signals. Similar issues arise in Figure 2, where blue cross-projections convey the signals $\mathbf{y}$ for use in both (3) and (17).

## C.4 Multilayer mirroring

In Figure 1b and accompanying text, we assumed that just one forward layer $\mathbf{y}_l$ was noisy, and just one feedback array $\mathbf{B}_{l+1}$ was adjusted, at any one time. Why not adjust all the $\mathbf{B}$'s at once? The problem is, when we adjust $\mathbf{B}_{l+1}$ we need a zero-mean (or $\beta$-mean), uncorrelated, equal-variance signal $\mathbf{y}_l$, which drives $\mathbf{y}_{l+1}$. But the resulting $\mathbf{y}_{l+1}$ generally will not be zero-mean, uncorrelated, or equal-variance, and so may not be effective at driving $\mathbf{B}_{l+2}$ toward $\mathbf{W}_{l+2}^T$. We can of course apply noise simultaneously to every *second* layer — for instance to $\mathbf{y}_2, \mathbf{y}_4, \mathbf{y}_6$, etc. — and adjust as many as half the $\mathbf{B}$'s at any one moment, so the mirroring does not really have to proceed one layer at a time. And there may be other options in networks with batch normalization or synaptic scaling [19]. Those mechanisms tend to keep all the forward signals approximately zero-mean and equal-variance (though not uncorrelated), and in that case it may be possible to adjust all the $\mathbf{B}$'s at once, driving the entire network with a single noisy input vector $\mathbf{y}_l$, though we haven't tested that idea here.

# D    Experimental details

## D.1    Architecture and training

We ran our experiments using 18- and 50-layer deep-residual networks ResNet-18 and ResNet-50 [30]. These networks consisted of sequences of sub-blocks, each made up of two (ResNet-18) or three (ResNet-50) convolutional layers in series. In parallel with these layers was a shortcut connection

whose output was added to the output from the convolutional layers. The output of the network passed through a final fully-connected layer followed by a softmax.

For the sign-symmetry algorithm, we carried out grid searches of the learning rate over the range [0.01, 2.0] while training out to 140 epochs to ensure convergence. We found that 0.5 gave the lowest top-1 errors with both ResNet-18 and ResNet-50, and so we used that value for all sign-symmetry experiments. Otherwise, all hyperparameters in all algorithms (except those for mirror mode) were taken from [31], including forward-path Nesterov momentum [32] 0.9 and a weight decay factor (L2 regularizer) $\lambda$ of $10^{-4}$. We used TF-Replicator [31] to distribute training across 32 TPU v2 workers, for a total minibatch of 2048 images. And we applied the annealing schedule from [31], i.e. $\eta_W$ grew linearly over the first 6 epochs (or over epochs 3 to 8 for weight-mirror networks, see below), and shrank 10-fold after epochs 32, 62, and 82.

## D.2  Mirroring

Each weight-mirror network spent its first two epochs entirely in mirror mode, bringing its initial, random weights into alignment. Thereafter, it did a small amount of mirroring after each minibatch of engaged-mode learning. It mirrored layer-wise: it created a new minibatch of noisy activity in layer $l$ (independent Gaussian signals with zero mean and unit variance across the 2048 examples in the mirroring minibatch) and it sent those signals through the convolutional layer and then the ReLU function. It computed the means of the post-ReLU outputs across the minibatch and subtracted them to give zero-mean outputs in each layer.

As in equation (21), the covariance matrix of these zero-mean signals was estimated by multiplying them and averaging over the minibatch. In convolutional layers, because of weight sharing, each weight connected multiple sets of inputs and outputs, and so we estimated the covariance associated with any given weight by averaging the estimated covariances over the pairs of inputs and outputs it connected.

We used these covariance estimates to train the feedback weights, as in (21), with a learning-rate factor $\eta_B$ of 0.1 and a weight decay $\lambda_{WM}$ of 0.5.

## D.3  Batch normalization

The weight mirror and Kolen-Pollack learned to match feedback matrices $\mathbf{B}_l$ to forward matrices $\mathbf{W}_l$, but didn't try to reproduce the batch normalization parameter vectors $\boldsymbol{\mu}$ or $\boldsymbol{\sigma}$ used in the forward path. In fact no $\mathbf{B}_l$ could have mirrored the combined effects of $\mathbf{W}_l$, $\boldsymbol{\mu}$, and $\boldsymbol{\sigma}$ in our convolutional networks, because the $\mathbf{B}_l$ matrices had the same convolutional, weight-sharing structure as the $\mathbf{W}_l$'s did — a structure which $\boldsymbol{\mu}$ and $\boldsymbol{\sigma}$ ignored. Therefore we simply passed the scaling parameter $\boldsymbol{\sigma}$ from the forward to the feedback path ($\boldsymbol{\mu}$ was not needed). This transfer involved very little information — just one scalar variable per feedback neuron — and could be avoided if we replaced convolution by more biological local connections *without* weight sharing.

In most of our experiments we applied batch normalization after the activation function, but the sign-symmetry method learned slightly better the other way, with normalization applied *before* the activation, as in [30]. We gave the numerical results for both cases in Section 6, but here we provide also a figure of the best result achieved by sign-symmetry in our tests, its 32.6(6)% final error with the ResNet-50 architecture:

Figure 5: ResNet-50 ImageNet results with batchnorm applied before ReLU.

# E  Pseudocode

---
**Algorithm 1** Weight Mirrors
---
1: **procedure** WEIGHT MIRRORS($network$, $data$)
    $\triangleright\ network$ has $L$ layers
2:    **for each** epoch **do**
3:       **for each** batch = $(\mathbf{y}_0,\ \mathbf{y}^*) \in$ data **do**
       $\triangleright$ Engaged mode
4:          compute the batch prediction $\mathbf{y}_L$                                           $\triangleright$ forward pass
5:          $\boldsymbol{\delta}_L = \mathbf{y}_L - \mathbf{y}^*$                                       $\triangleright$ compute error
6:          **for** layer $l$ **from** $L$-1 **to** 0 **do**
7:             $\mathbf{W}_{l+1} = (1-\lambda)\,\mathbf{W}_{l+1} - \eta_W\,\boldsymbol{\delta}_{l+1}\,\mathbf{y}_l^T$     $\triangleright$ equation (4), with weight decay
8:             $\mathbf{b}_{l+1} = (1-\lambda)\,\mathbf{b}_{l+1} - \eta_W\,\boldsymbol{\delta}_{l+1}$
9:             $\boldsymbol{\delta}_l = \phi'(\mathbf{y}_l)\,\mathbf{B}_{l+1}\,\boldsymbol{\delta}_{l+1}$         $\triangleright$ compute error gradients using $\mathbf{B}$, equation (3)
10:         **end for**
       $\triangleright$ Mirror mode
11:          **for** layer $l$ **from** 1 **to** $L$-1 **do**
12:             sample  $\mathbf{y}_l \sim \mathcal{N}(\mu,\ \sigma^2)$                $\triangleright$ ideally zero-mean, small-variance
13:             $\mathbf{y}_{l+1} = \phi(\mathbf{W}_l\,\mathbf{y}_l + \mathbf{b}_l)$
14:             $\boldsymbol{\delta}_l = \mathbf{y}_l - \bar{\mathbf{y}}_l$                           $\triangleright$ subtract batch average
15:             $\boldsymbol{\delta}_{l+1} = \mathbf{y}_{l+1} - \bar{\mathbf{y}}_{l+1}$         $\triangleright$ forward cells drive feedback cells
16:             $\mathbf{B}_{l+1} = (1-\lambda_{WM})\mathbf{B}_{l+1} + \eta_B\,\boldsymbol{\delta}_l\,\boldsymbol{\delta}_{l+1}^T$       $\triangleright$ equations (7) and (21)
17:          **end for**
18:       **end for**
19:    **end for**
20: **end procedure**

---

---
**Algorithm 2** Kolen-Pollack algorithm
---

1: **procedure** KOLEN-POLLACK ALGORITHM($network$, $data$)
  ▷ $network$ has $L$ layers
2:     **for each** epoch **do**
3:         **for each** batch = $(\mathbf{y}_0,\ \mathbf{y}^*) \in$ data **do**
4:             compute the batch prediction $\mathbf{y}_L$             ▷ forward pass
5:             $\boldsymbol{\delta}_L = \mathbf{y}_L - \mathbf{y}^*$             ▷ compute error
6:             **for** layer $l$ **from** $L$-1 **to** 0 **do**
7:                 $\mathbf{W}_{l+1} = (1-\lambda)\,\mathbf{W}_{l+1} - \eta_W\,\boldsymbol{\delta}_{l+1}\,\mathbf{y}_l^T$     ▷ equation (16)
8:                 $\mathbf{b}_{l+1} = (1-\lambda)\,\mathbf{b}_{l+1} - \eta_W\,\boldsymbol{\delta}_{l+1}$
9:                 $\mathbf{B}_{l+1} = (1-\lambda)\,\mathbf{B}_{l+1} - \eta_B\,\mathbf{y}_l\,\boldsymbol{\delta}_{l+1}^T$     ▷ equation (17)
10:                $\boldsymbol{\delta}_l = \phi'(\mathbf{y}_l)\,\mathbf{B}_{l+1}\,\boldsymbol{\delta}_{l+1}$    ▷ compute error gradients using $\mathbf{B}$, equation (3)
11:             **end for**
12:         **end for**
13:     **end for**
14: **end procedure**