[Reviews · NeurIPS 2019]

Reviewer 1



With both methods, they demonstrate similar performance to backpropagation on ResNet-18 and ResNet-50 architectures on ImageNet. To me, this is the biggest strength of the paper. There have been many proposals for learning algorithms that do not rely on weight transport (feedback alignment included), which have only been evaluated on toy tasks and on non-convolutional neural networks. The only weakness I would say is if there were experiments with more architectures on ImageNet using their learning algorithm, either variants of ResNet (101, 152, etc) and outside of the ResNet family (VGG, AlexNet, EfficientNets). If that is difficult to do, is it because the hyperparameters to get this algorithm to work are more than those for standard backpropagation? If that is indeed the case, that should be made explicit in the main text of the paper.

Reviewer 2



The study proposes two alternative mechanisms that do not need weight transport as used in backprop to mimic biologically plausible learning. The authors introduce the two mechanisms, mirror weight and Kollen-Pollack (KP) algorithm, in an way that is easy to understand, while providing sufficient mathematical details. The experiments show that, both methods, particularly the KP one, provide lower test error on imagenet task with smaller resnet, compared to feedback alignment, sign-symmetry and backprop, while with larger resnet feedback alignment clearly wins. Both weight mirroring and KP were found to be successful in terms of aligning the forward and backward weight matrices. Although the results are preliminary and more experiments are definitely needed to establish these alternative mechanisms, I find this a nice piece of work, which demands attention and further investigation from the community. Minor comments: On line 168, it was mentioned that sign-symmetry did better in other experiments - but they were not shown. I would encourage authors to include additional experiments in SI.

Reviewer 3



Getting rid of weight transport is an important step towards biologically plausible deep learning algorithms and both proposed mechanisms seem to be valid contributions towards this goal. While not entirely surprising, the empirical results show the effectiveness of the proposed methods. The biggest criticism to be brought forward is perhaps the fact that both methods rely on one-to-one mappings between y and delta nodes. This assumption can probably hardly be met in biological systems, and it is not obvious how this can be circumvented. It would be good if the authors could comment on the plausibility of this assumption and possibly discuss alternative scenarios (e.g. is there a way to have a linear mapping between delta and y, rather than the identity? Are there known circuits which somewhat resemble this direct correspondence?) -- The authors have addressed my concerns in the rebuttal. I'm updating my score.

[Author Response · NeurIPS 2019]

The reviewers have raised five issues:

### Testing other architectures

The most interesting architectures, in our view, are locally connected but *non*-convolutional networks, like those in the brain. We are working on tests of this type, but there are technical hurdles, as large non-convolutional networks are still extremely slow on computers. We will also investigate training AlexNet and VGG architectures using our approaches.

### Figures of experiments where Sign-Symmetry achieved its best results

We will show these plots in a new Appendix.

### Alignment

In Figure 4, KP kept matrix and $\delta$ angles smaller than WM did, but that may not be the case in all learning tasks. With KP, $\mathbf{B}$ converges to $\mathbf{W}^T$ at a rate that depends on $\lambda$, the weight-decay factor in equation (17). A big $\lambda$ speeds up alignment, but may hamper learning. So the question is whether we can find a good balance between weight decay $\lambda$ and learning rate $\eta_W$, but at present we have no mathematical proof that a good balance will always be possible. In this respect, WM may be more versatile than KP, because if mirroring ever fails to yield small enough angles, we can simply do more mirroring, e.g. in sleep.

### Computational costs

Suppose layers $l$ and $l+1$ are fully connected, with $n_l$ and $n_{l+1}$ forward units (and the same numbers of feedback units if they are separate from the forward ones), and let $n = min(n_l, n_{l+1})$. Then for each training example, KP does $n + 4n_l n_{l+1}$ flops to adjust $\mathbf{B}_{l+1}$ using equation (17). WM does the same number to adjust $\mathbf{B}_{l+1}$ using equation (7) and weight decay. But WM also has to generate a random vector $y_l$ and then perform about $2n_l n_{l+1}$ flops to compute $y_{l+1}$ from $y_l$ using equation (1), whereas KP uses the same $y_l$ and $y_{l+1}$ that train the forward matrices. In short, WM needs twice as many forward passes as KP does to collect as many training examples for its $\mathbf{B}$ matrices (whether the net is fully-connected or not).

In our ResNet-18 and ResNet-50 tests, the computational costs of WM's additional forward passes were 1.8 GFLOPs and 3.8 GFLOPs respectively, not counting the costs of random number generation.

### Could the brain have one-to-one wiring between forward and feedback neurons?

Getting that one-to-one correspondence is of course trivial if the *same* neurons make up the forward and feedback paths, though then we face the new problem of signal segregation — explaining how signals $\mathbf{y}$ and $\delta$ can flow through the same cells without interfering. Some possibilities are that neurons segregate $\mathbf{y}$ and $\delta$ by conveying them with different intracellular messengers or computing them in different parts of the cell [22,29], or by multiplexing [23], or by taking turns carrying one or the other signal.

If the forward and feedback paths are *distinct* sets of neurons, then the one-to-one connections might arise during development. We know that very precise and consistent neuronal wiring is found in simple organisms such as C. elegans and in the compound eyes of insects, while in primate cerebellum there is a mechanism (not fully understood) that wires up each Purkinje cell with exactly one appropriate climbing fiber.

And finally, something less than strict one-to-one wiring may suffice for effective learning. As in the case of weight transposes, an approximate one-to-one wiring might be achieved by simple local learning rules.

We agree with the reviewer that this question is important, and we now address each of these options for circuitry arrangement in an extended paragraph in the updated manuscript.

*References:*

[1]-[28] In main text.

[29] BA Richards, TP Lillicrap, *Dendritic solutions to the credit assignment problem*, Current opinion in neurobiology 54, 28-36.


[Meta-Review · NeurIPS 2019]

The reviewers agree that the paper has strong theoretical, algorithmic contributions that are well evaluated and analyzed. Clear accept.